# Quercetin and Its Nano-Formulations for Brain Tumor Therapy—Current Developments and Future Perspectives for Paediatric Studies

**DOI:** 10.3390/pharmaceutics15030963

**Published:** 2023-03-16

**Authors:** Aida Loshaj Shala, Ilaria Arduino, Mimoza Basholli Salihu, Nunzio Denora

**Affiliations:** 1Department of Drug Analysis and Pharmaceutical Technology, Faculty of Medicine, University of Prishtina, 10000 Prishtina, Kosovo; 2Department of Pharmacy—Pharmaceutical Sciences, University of Bari “Aldo Moro”, Orabona St. 4, 70125 Bari, Italy

**Keywords:** paediatric brain tumors, anti-cancer, quercetin, nano-formulations

## Abstract

The development of efficient treatments for tumors affecting the central nervous system (CNS) remains an open challenge. Particularly, gliomas are the most malignant and lethal form of brain tumors in adults, causing death in patients just over 6 months after diagnosis without treatment. The current treatment protocol consists of surgery, followed using synthetic drugs and radiation. However, the efficacy of these protocols is associated with side effects, poor prognosis and with a median survival of fewer than two years. Recently, many studies were focused on applying plant-derived products to manage various diseases, including brain cancers. Quercetin is a bioactive compound derived from various fruits and vegetables (asparagus, apples, berries, cherries, onions and red leaf lettuce). Numerous in vivo and in vitro studies highlighted that quercetin through multitargeted molecular mechanisms (apoptosis, necrosis, anti-proliferative activity and suppression of tumor invasion and migration) effectively reduces the progression of tumor cells. This review aims to summarize current developments and recent advances of quercetin’s anticancer potential in brain tumors. Since all reported studies demonstrating the anti-cancer potential of quercetin were conducted using adult models, it is suggested to expand further research in the field of paediatrics. This could offer new perspectives on brain cancer treatment for paediatric patients.

## 1. Introduction

Brain tumors are a heterogeneous group of cancers with a high mortality rate [1]. While brain tumors are not uncommon in adults, they are the most common solid tumors in children and a major cause of illness and death among young people [2]. Paediatric central nervous system (CNS) tumors are the second most prevalent paediatric malignancy and the most common solid tumor in children. They continue to be the largest cause of cancer-related mortality in children aged 0–14 and improved therapies have long been sought [3,4,5]. 

The most frequent brain tumor in children are gliomas, representing nearly 50% of all paediatric CNS tumors [6]. Gliomas are tumors that evolve from CNS-supporting cells known as glial cells. Based on implicated cell type, gliomas are classified into several forms, such as astrocytoma, glioblastoma, ependymoma, oligodendroglioma, and oligoastrocytoma. The most frequent brain tumors in paediatric populations are glioblastoma, medulloblastoma, and ependymoma [7,8]. 

The standard treatment protocol consists of surgery, followed by chemotherapy (mostly temozolomide) and radiation. The efficacy of this conventional treatment is extremely limited, and it is linked with a poor prognosis, with a median survival of fewer than two years [9].

Although several treatment strategies for brain cancer have been developed, there are only a few drugs that have been clinically approved for medical use [10]. Consequently, the need for the advancement of effective treatment options is significant and urgent [11]. 

Various studies have indicated an increase in the utilization of plant-derived products to manage various diseases, including brain tumors [12]. Therefore, intending to improve therapeutic options for CNS malignancies, important efforts have been made to investigate novel natural alternative therapies that are warranted for use alone or in combination with other pharmacological agents [13,14]. 

Among them, one of the most important naturally occurring organic compounds are polyphenolic flavonoids. They are most commonly derived from plants, fruits and vegetables and are considered natural and non-toxic chemopreventers. These compounds demonstrate remarkable antioxidant, anti-inflammatory and anticancer properties [14,15,16].

Quercetin (Que), also known as 3,3′,4′,5,7-pentahydroxyflavone (Figure 1), is one of the main flavonoids found in human diets with daily consumption that ranges from 3 to 38 mg [17,18,19]. The highest sources of Que include asparagus, apples, berries, cherries, onions and red leaf lettuce whereas the lowest sources are green peppers, broccoli, tomatoes and peas [20]. Que exhibit several biopharmacological activities including antioxidant, anti-inflammatory, antidiabetic, antifungal, antiobesity, antiviral, antibacterial and neuroprotective activities [21,22].

Que, as one of the main representatives of the polyphenols with outstanding anticancer and antioxidant activity, has attracted the attention on the research field of cancer biology and cancer treatments. Extensive research evidence involves Que in numerous tumor-related activities such as oxidative stress, proliferation, apoptosis, angiogenesis, tumor necrosis factor, cell cycle and metastasis [23,24,25,26,27].

Studies have shown that Que can induce apoptosis by activating caspase-3 and caspase-9, and downregulating anti-apoptotic Bcl-2 proteins [28,29,30]. It also inhibits cell proliferation by regulating the cell cycle and inducing G1 arrest [31]. Que’s anti-angiogenic effects are mediated through the inhibition of vascular endothelial growth factor (VEGF) and its receptor, which are important for the growth of new blood vessels in tumors [32]. Additionally, Que has been shown to enhance the efficacy of chemotherapy and radiotherapy in brain tumors by increasing the sensitivity of cancer cells to these treatments [14,23,33]. 

Based on preclinical research studies, the reported data suggests the potential anticancer activity of Que in brain tumors. The most commonly employed testing models for these findings are adult glioma cell lines. Although gliomas and glioblastomas are prevalent tumors in children, none of the studies performed so far have utilized paediatric cell models. Therefore, the objective of this review is initially to assemble the potential activity of Que in brain tumors by examining existing studies on Que’s anticancer activity and its mechanisms. Additionally, we emphasize the bioavailability limitations that restrict its usage and highlight the benefits of Que nano-formulations that can overcome these limitations. Finally, we propose further investigations into the anti-cancer activity of Que in the field of paediatrics, considering and discussing significant barriers and potential benefits that may result in novel insights into the treatment of brain cancer in children.

## 2. Quercetin’s Effects in Brain Tumors: Molecular Mechanisms and Signaling Pathways

According to numerous studies (presented in Table 1), Que anticancer properties are linked to the targeting of a large variety of cellular processes, with a focus on apoptosis, autophagy and metastasis (Figure 2) [34].

The process of “programmed cell death”, or apoptosis, is brought on by either internal signals (such as genotoxic stress) or external signals (binding of cellular ligands to death receptors) [35]. 

Apoptosis begins as a result of these signals which promote a long path of activation of different enzymes, ending with cellular destruction and the removal of damaged cells [35]. A combination of intrinsic and extrinsic mechanisms that start the molecular cascade of apoptosis, activate caspases −3/−6/−7, which carry out the destruction of vital cellular substrates [35,36]. Therefore, it is assumed that apoptosis is a process that is monitored and carried out by a range of molecules, and their improper functioning in issues results with avoidance of apoptosis which is crucial in the development of the tumor. As a result, diverse therapeutic drugs targeting the apoptotic process may represent an effective alternative for the treatment of cancer patients [36].

According to Tavana and his team, Que causes apoptosis in a variety of glioblastoma cell lines, such as C6, U87, U138, U373, and GL-261 [29]. After the treatment of GBM cells (U87-MG, U251, SHG44 cell lines) with Que, a low level of the anti-apoptotic protein Bcl-2 was detected. In addition, PI3K/AKT and Ras/MAPK signaling pathways were both inhibited [24]. Furthermore, it was observed that Que inhibits the extracellular expression of fibronectin and MMP-9 matrix proteins, which are associated with cell invasion and migration [24]. 

On the other hand, it was reported that Que pro-oxidant impact urged apoptosis [37]. It has been shown that Que simultaneously exhibits antioxidant and pro-oxidant effects in a variety of cell lines. Determination of the effect that is prevalent varies from the concentration of Que that was used and duration of the exposure [38]. On the U138MG human glioma cell line, necrotic and apoptotic cell death were also noted, but with a fundamental difference. Initially, after 24 h, Que treatment caused cell necrosis, while apoptosis was observed with higher doses and after 48 h. The authors suggested that Que interaction with mitochondrial membrane decreased ATP levels, which led to cell necrosis [39]. However, it is worth noting that apoptosis as a programmed cell death is more favorable for tumor damage, in comparison to necrosis which is followed by inflammatory response and stimulation of damaging molecular pathways. Necrosis is also associated with a low prognosis of the patient and increased tumor malignancy [27].

One other study reported that after the treatment of A172 glioblastoma cell line with Que, it was observed that cell death was associated with caspase-dependent apoptotic pathways rather than the production of reactive oxygen species (ROS) [40]. It was suggested that apoptotic cell death resulted by the downregulation of extracellular-signal-regulated kinase (ERK) pathway (which is involved in functions including the regulation of meiosis, mitosis and postmitotic functions in differentiated cells), Protein Kinase B—Akt (anti-apoptotic functions) and survivin (antiapoptotic protein) [40].

Apoptotic cell death was also investigated in the U373MG cell line, where treatment with Que induced the transcription of independent p53 apoptotic pathway. It was demonstrated that through the activation of c-Jun N-terminal kinase (JNK) pathway, the levels of p53 protein are elevated. These increased levels of p53 promote apoptosis through the above-mentioned pathway [31].

Additionally, Que involvement in the inhibition of proliferation in glioma cell lines was also observed. It was demonstrated that Que is involved in the down-regulation of ecto-5′-NT/CD73 enzyme which produces adenosine that has a crucial role in proliferation, cytoprotective effect and ATP production for these cells. Therefore, by decreasing ecto-5′-NT/CD73 function, Que interferes in vital cellular functions of glioma cells, leading to inhibition of proliferation, tumor migration and invasion [41].

Involvement of Que in inhibition of proliferation in glioma cells was also demonstrated from Park et al. in 2011. Que decreased the level of Phospholipase D1 (PLD-1) enzyme that is associated with proliferation and suppression of apoptosis in tumor cells. From the obtained results, it has been shown that Que decreased the expression of PLD-1 at a transcriptional level by inhibiting NF-κβ transactivation. PLD-1 produces phosphatidic acid (PA), which is associated with the activation of matrix metalloproteinase-2 (MMP-2), i.e., extracellular matrix proteins directly involved in tumor metastasis. Therefore, by decreasing the level of PLD-1, Que might reduce the tumor activity through the reduction of proliferation and invasion of gliomas [42]. 

The results obtained from these studies suggest that Que has a multifaceted anti-cancer potential and mechanisms that slow down tumor growth. Each study approached the investigation of Que anticancer effects on these cells differently, targeting various pathways, molecular targets, and processes. However, despite the extensive research conducted so far, a clear understanding of the multitargeted molecular mechanisms highlighted in preclinical research still requires further investigation. 

Although gliomas and glioblastomas are prevalent tumors in children, none of the studies mentioned above employed paediatric cell models. Adult gliomas were examined instead, due to limited research material availability and a lack of literature in the paediatric field. Therefore, there is a need for further research, starting with in vitro studies employing paediatric cellular models, followed by in vivo investigations and clinical trials.

**Table 1 pharmaceutics-15-00963-t001:** An overview of some studies on quercetin’s mechanism of action in tumor cell lines.

Type of Que	Cell Line	Results	Conclusions	References
Que	U87-MG U251 SHG44	↓ P-Akt ↓ P-ERK ↓ Bcl-2 ↓ MMP-9 ↓ Fibronectin	- inhibition of viability and migration of glioma cells - Apoptosis induction - Suppression of RAS/MAPK/ERK and PI3K/Akt pathways	Pan et al., 2015 [24]
PEG2000-DPSE-coated quercetin nano-particles	C6	↑ ROS production ↑ Cytotoxicity ↑ P53 expression ↓ Membrane potential - Cytochrome c release - Caspase activation	PEG2000-DPSE-coated quercetin nano-particles showed: - dose dependent toxicity on C6 cells - apoptosis induction through ROS accumulation - Enhancement of the anticancer effect of quercetin on C6 cells	Wang et al., 2013 [37]
Que	A172 LBC3	↑ ROS generation ↑ Caspase 3/7/9 activity ↑ CHOP - Deregulated SOD1 and SOD2 expression ↓ ATP	- induction of apoptosis via intrinsic pathway on A172 cell line - apoptosis/necrosis shift after 48 h of treatment on LBC3 cell line	Kusaczuk et al., 2022 [27]
Que	U138MG	↓ cell membrane permeability ↑ Caspase 3/7 activity after 48–72 h of treatment - No induction of caspase activation at 30 µmol/L - accumulation of cells at G2 phase ↓ mitotic index - QUE (100 µmol/L) did not cause hippocampal organotypic culture damage - the neuronal injury was reduced to 35% after QUE treatment	↓ Cell proliferation - Necrotic and apoptotic cell death - Arrest at the G2 checkpoint of the cell cycle - Neuroprotective effects	Braganhol et al., 2006 [39]
Que	A172	↓ Cell viability - No ROS generation - down-regulation of ERK and Akt ↓ mitochondrial membrane potential ↑ caspase 3-activity ↓ survivin expression	- Apoptosis induced by QUE through caspase-dependent mechanisms - Involvement of Akt, ERK and survivin downregulation - ROS generation is not the cause of apoptosis	Kim et al., 2008 [40]
Que	U373MG	↑ cells in the sub-G1 phase ↓mitochondrial membrane potential ↑ Caspase 3/7/9 activation and activity - Activation of JNK ↑ p53 expression ↑ AVO	- QUE induces apoptosis through increase p53 expression - QUE induces protective autophagy - QUE combined with an autophagy inhibitor can be an effective anticancer agent	Kim et al., 2013 [31]
Que	U87	↓ PLD1 expression - inhibition of PLD1 activity by QUE - inhibition of NFκB transactivation ↓ MMP-2 activity	- Through PLD-1 suppression by QUE, glioma cell proliferation is achieved - QUE suppressed PLD1 expression through inhibition of NFkB transactivation - QUE inhibited tumor cell invasion by reducing MMP-2 activity.	Park et al., 2011 [42]
Que	U138MG	↑ concentrations of adenine nucleotide/ nucleoside in the extracellular medium - ecto-5′-nucleotidase/CD73 converts AMP to adenosine - QUE inhibits ecto-5′-nucleotidase/CD73 - U138MG proliferation was inhibited by an ecto-5′-nucleotidase/CD73 inhibitor (APCP)	- Inhibition of ecto-5′-NT/CD73 may slow the tumor proliferation by reducing the synthesis of extracellular adenosine.	Braganhol et al., 2007 [41]

MMP-9, matrix metallopetidase; AKT, Protein kinase B; ERK, extracellular signal-regulated ki-nase; U87-MG, glioblastoma cell lines; U251, glioma cell lines; SHG44 glioma cell lines; Bcl-2, B-cell leukemia/limphoma 2 protein; ROS, Reactive Oxygen Species; P53; p53, tumor protein; A172, glioblastoma cell lines; LBC3, glioblastoma cell lines; CHOP, C/EBP homologous protein; PLD1, Phospholipase D1; ↑ Increase; ↓ Decrease.

## 3. Bioavailability and Metabolism of Que

Bioavailability is considered as a fraction of an orally administered drug that is absorbed and accessible for physiologic activity or storage [43]. Based on pharmacokinetics context, bioavailability is classified as absolute or relative. Absolute bioavailability is more accurate, and presents the fraction of administered dose that reaches systemic circulation (comparison between drug exposure via extravascular and intravenous administration of the tested dosage form) [44].

Otherwise, relative bioavailability is simpler, but less accurate. The estimation of relative bioavailability includes a comparison of two formulations or two routes of administration of the same formulation without involving intravenous administration. However, studies examining the relative bioavailability of flavonoids in general are more common [45,46,47,48,49,50]. 

Que, like most flavonoids, has very low bioavailability, which significantly reduces its applications as a therapeutic agent [49]. Low hydrophilic property and high lipophilicity of Que determine its low bioavailability after ingestion. The small amount of absorbed Que reaches circulation and remains available for distribution in peripheral tissues [51,52]. Absorption pathways of Que via gastrointestinal tract of humans and other mammal are quite well understood [53].

As it is well documented, the primary site of Que absorption is the small intestine [54], while a minor proportion is absorbed in the stomach [55,56]. Presentation of Que bioavailability after ingestion of aglucone and glycoside forms is presented in Figure 3.

To date, a large number of studies have been conducted in order to establish and confirm the bioavailability of Que that was previously described.

As it is shown, Que aglycone and glycoside forms go through different routes of digestion and absorption [23,24,25]. Unmetabolized lipophilic aglycon passively diffuses from the intestinal lumen into the enterocyte where it is directly absorbed into the hepatic portal vein or metabolized before absorption. It has been shown that the main metabolic pathway of Que involves phase I metabolism (oxidation and O-demethylation) and phase II metabolism (glucuronidation, sulfation and methylation). Glucuronidation, which is the principal phase II metabolism, is provided through uridine diphosphate-glucuronic acid transferase (UGT) in gastrointestinal cells, sulfation by sulfotransferases (SULTs), and/or methylation by catechol-O-methyl transferase (COMT) present in intestinal and hepatic cells [53,57,58,59]. The same metabolic outcomes are also observed in vitro in rat or human hepatocytes [60]. These conjugated forms of Que are present in bloodstream circulation [61,62].

The conjugated Que is more stable, although with decreased biological activity compared with aglycone [63,64,65,66].

Furthermore, it is worth noting that absorption of Que in glucoside form (i.e., Que 3-O-glucoside, i.e., isoquercitrin) into circulation is higher than the unglycosylated Que or its other glycosides, e.g., rutinoside (rutin) or galactoside [51]. On the intestinal brush border, lactase-phlorizin hydrolase enzyme (LPH) hydrolyzes glycosides to aglycones. Alternatively, the glycosides form of Que can be transported by sodium-dependent glucose transporter 1, (SGLT-1) which can actively transport some glucosides through the membrane of enterocytes where they are hydrolyzed by cytosolic ß-glucosidase [67,68]. Following absorption, the remaining aglucone and metabolites of Que using serum albumin are transported to the liver [69]. In the liver, the metabolites and the remaining Que in aglucon form undergo phase I and II metabolism, and the resulting metabolites which are transported along with intestinal metabolites enter the systemic circulation for distribution into body tissues [70,71].

In 2003, Khaled et al., using the rat animal model, investigated the pharmacokinetics and mean time tissue distribution parameters of Que after the I.V and oral (solution and suspension) administration. They reported that the absolute bioavailability from aqueous solution was 16%, whereas, for ethanol-PEG, it reached 27.5% [72,73].

Later, Simioni and his team in 2018, reported about Que absorption in healthy volunteers. From the obtained results, they reported that after the administration of 100 mg/kg the portion of absorbed Que was between 3% and 17% [74]. Previously, Xiao et al., in 2005, after gavage administration of radio labelled Que in rats, reported that only 20% of total Que was absorbed and it was found as metabolized or in free form [75].

Additionally, it was observed that after a single oral administration of 10 mg Que per 200 g of the body weight, 93% of Que was metabolized after an hour [76].

Some studies have shown that Que metabolites pass the blood–brain barrier (BBB) reaching sufficient amounts in the brain tissue necessary to exert their pharmacological activity [77]. However, it has been demonstrated on in vivo studies that effective doses of Que activity in CNS disorders were relatively high. In general, for substances with low bioavailability, in order to achieve the pharmacological activity, high doses are required [68]. 

Therefore, enhancement of systemic bioavailability includes various methods that increase solubility. Among them, the most applied so far are reduction of particle size (nano-particles, nano-suspensions) and chemical modification (derivatization, complexation). Thus, nano-technology may be an excellent tool to increase solubility, since nano-carriers loaded with Que increase the time of circulation, and passage through biological barriers and avoid renal clearance and the immune system, leading to an increment of the bioavailability.

## 4. Nano-Delivery Systems Entrapping Que—New Approaches for Bioavailability Improvement

Different formulations of Que have been developed to improve bioavailability profile. Biedrawa et al. in 2022 reported that based on a survey that they performed from different databases, during the period 2012 to 2022, Que-loaded nano-particles presented the most investigated novel formulations of Que in terms of neurodegenerative disorders (61.9%), followed by nano-lipid carriers and exosomes (9.52%), while nano-capsules, microparticles, nano-emulsions and nanofibers are less investigated and analyzed with only 4.76% [68]. The main focus of nano-delivery systems is an increment of bioavailability which comes as a result of the avoidance of limitations such as solubility and dissolution profile in the gastrointestinal system [78]. Additionally, in brain tumor treatments, the major challenge is to achieve targeted delivery of an active compound by passing through BBB [79]. To overcome the obstacles provided by the BBB, it is essential to explore the changes affecting it, to understand how to exploit these findings in the study and design of innovative formulations targeting the brain. It is also very important to exploit the concept of age-related targeting, as it allows the type of treatment to be considered according to different needs and peculiarities depending on the disease and age of onset [80].

To overcome the bioavailability limitations of Que in brain tissue and enhance target specific action in tumor cells, some of the recent novel Que formulations will be further discussed (presented in Figure 4 and summarized in Table 2).

Recently, very promising results were reported using exosome as a nano-carrier for Que, aiming to enhance its accumulation in the brain. Exosomes are nano-scale sized vesicles secreted by living cells. Good biocompatibility, low immunogenicity and high level of transmission are some exosome advantages compared with other nano drug carriers. Exosomes possess the innate ability to cross through BBB [81]. These characteristics of exosomes are also confirmed by huge number of studies that have shown a unique role of an exosome not only in drug bioavailability improvement but also in BBB penetration, leading to the achievement of drugs targeting the brain [82]. In this study, plasma exosomes were developed and loaded with Que. These formulations were administered i.v. in rats for further pharmacokinetic studies. Based on the reported results, pharmacokinetic parameters were better in Que-exosome delivery system compared to Que alone (half-life 89.14 min vs. 36.47 min, maximum plasma concentration 2.5 mg/L vs. 1.31 mg). Moreover, the authors reported that exosomes increased the concentration of Que in the brain up to 2.5-fold in the cerebrum and 1.5-fold in cerebellum compared with free Que. These findings are in good accordance with previous reports, that exosomes as a nano-carrier enhanced brain targeting of the incorporated compound [83].

It is widely documented in the literature that solid lipid nano-particles (SLNs) [84] and nano-structured lipid carriers (NLCs) represent nano-particle systems with an innate ability to overcome the BBB even without any functionalization [85], it is also possible to functionalize the particle outer shell with several ligands. The addition of targeting moiety on the surface of these lipid nano-vectors allows them to be directed towards a specific target and to interact with molecules on the target tissue. This modification is able to increase and improve the uptake of the nano-systems [86,87,88].

Recently, Pinheiro et al. in 2020 developed lipid nano-particles loaded with Que and functionalized them with RVG29 peptide [89]. Their study was based on previous data, suggesting that functionalized nano-particles with specific ligands can target specific cells or respond to various stimuli in the target site, making them unique carriers for brain delivery [90,91]. One of the most expressed receptors in pre and post-synaptic sites of neuron cells and in the brain are endothelial cells that are present in the BBB are nicotinic acetylcholine receptors (nAChR) [92,93,94]. The peptide RVG29 is a fragment of 29 amino acids from the rabies virus glycoprotein, which can interact with these receptors [95,96]. This peptide was used for the functionalization of nano-particles in order to increase brain delivery. This is the first reported study that used RVG29 peptide for Que nano-delivery into the brain. Initially, lipid nano-particles (solid lipid nano-particles—SLN and nano-structured lipid carriers—NLC) were prepared and functionalized with RVG29. The RVG29 functionalization was confirmed with NMR and FTIR spectroscopic techniques. After the characterization analysis (TEM images, zeta potential), it was warranted that the properties granted good stability, with no aggregation and also assured the appropriate characteristics for brain delivery. The encapsulation efficiency resulted above 80%, indicating high Que encapsulation inside the nano-particles. This formulation after the tests performed in hCMEC/D3 cell line, which is a human BBB model due to its similarities to the real BBB, has shown the increase of 1.5-fold in the permeability across the in vitro model of BBB compared with the non-functionalized nano-particle. Moreover, after the in vitro LDH assay that was conducted for cytotoxicity and biocompatibility studies, the results revealed that they are safe; even with the highest concentration, the cytotoxicity is lower than 15% [89]. Based on the reported results, this study suggested that these nano-system deliveries can increase the permeability across the BBB.

In the same line with the above-explained study, the same team of researchers formulated Que-loaded lipid nano-particles and functionalized them with transferrin [97]. The same method for Que encapsulation in lipid nano-particles (SLN and NLC) and functionalization with transferrin were used as in the previously described study [89]. In order to facilitate the passage across the BBB, transferrin was used for the functionalization of lipid nano-particles, due to the overexpression of transferrin receptors in endothelial cells in the brain. Furthermore, the confirmation of functionalization of lipid nano-particles with transferrin using NMR and FTIR spectroscopy, was performed as well. The characterization of prepared Que nano-particles following with in vitro testing by hCMEC/D3 cell line, in order to evaluate the capacity of nano-particles to penetrate through BBB, has shown that NLC showed better permeability results compared to SLN, both for non-functionalized and functionalized nano-particles. However, from the tested results it was reported that nano-particles functionalized with transferrin did not reveal an increase in their permeability ability compared to non-functionalized nano-particles. The authors suggest that this might be a result of the transferrin receptors saturation. Additionally, a cytotoxicity test was conducted using LDH assay on hCMEC/D3 cells. No significant cytotoxic effect was observed, indicating that these nano-particles are safe at the range of concentrations that were tested [97].

Nano-particles, as one of the excellent tools of nano-technology, present an efficient vehicle for drug delivery into numerous tissues and organs including CNS. One of the most interesting and promising nano-particles are superparamagnetic iron oxide nano-particles (SPION, Fe_3_O_4_ NPs). These nano-particles could be used as a targeted drug delivery system due to the magnetic properties of the external magnet [26,98]. Based on the advantages of SPION as a carrier system, an interesting study reported how they develop Que-SPION and analyze their bioavailability in the brain of animal models [99]. The Que conjugated with dextran-coated iron oxide nano-particles were prepared with chemical precipitation. The authors used iron oxide nano-particles coated with dextran in order to reduce toxicity at high concentrations. After the characterization of all the necessary parameters, Que alone and Que-SPIONs were injected in healthy rat models for bioavailability studies. The plasma concentrations of Que in rats treated with QT- SPION were higher than those of free Que, while none of them remain in plasma for more than 9 h. Moreover, based on the reported results, it was found that the concentration of Que in the brain delivered by Fe_3_O_4_ nano-particles (50 mg/kg and 100 mg/kg) were about 7 and 10-fold higher, respectively, in comparison with the free Que. Therefore, the concentration of Que in plasma and brain are significantly higher due to conjugation with Fe_3_O_4_ nano-particles. Furthermore, in order to verify whether SPION crossed the BBB, the level of iron was measured in plasma and brain. The results of this study showed that SPION has the ability to penetrate from the digestive system to the blood circulation [99]. Still, the content of iron in the brain of treated rats is not significantly higher than in control rats. Therefore, this study suggests that SPION cannot cross BBB, which is in good coherence with some previously reported data [100].

Furthermore, the enhancement of bioavailability of Que using the SPION drug delivery system was also reported in one other study performed by Amanzadeh and his team in 2019 [101], which is in good accordance with previously described findings [99].

Additionally, one other investigation, developed carboxylate Que loaded in SPION. However, in contrast to the study mentioned earlier, SPION carriers were initially functionalized with APTES ((3-Aminopropyl) triethoxysilane) and polyethylene glycol (PEG). After SPION functionalization with APTES and PEG improvement, morphological, structural, spectroscopical and magnetic properties were observed. Afterwards, since folic acid (FA) is a specific ligand that is commonly over-expressed on the surface of many human cancers, functionalized SPION was conjugated with FA, which was used as a targeting agent for cancer cells. Carboxylate Que (CQ) was loaded as an anticancer drug. Finally, the cytotoxic effect of the carboxylate Que was tested on L929 fibroblast cells (folic acid receptor negative cells) and U87 glioblastoma cell lines using MTT assay. Based on the observed results, it was found that Que from SPION@APTES@FA-PEG@CQ nano-drug system was released under acidic conditions in vitro. Moreover, SPION@APTES@FA- PEG did not show a cytotoxic effect, since according to MTT assay, this formulation decreased the cell viability on U87 cell lines. Based on these observations, it was concluded that e SPION@APTES@FAPEG@ CQ nano-drug might have great potential for the targeted treatment of brain cancers [102]. Yi et al., in 2020, introduced an interesting strategy for the development of nano-composites (NC) as a drug delivery vehicle. They used a combination of Que and Na_2_SeO_3_ to obtain selenium (Se) nano-particles [103]. Selenium is a vitally necessary microelement, which usually in the human body can be found as a selenium protein and selenocysteine [104]. Based on the structural features of selenium protein as an organic form of selenium, they developed a method to produce Que-loaded selenium nano-particles (Que@Se NP). These nano-particles were coated with acacia and polysorbate 80 (P80) to form P80-Que@Se NC. P80 was used to enhance the permeability across BBB and acts as a pharmaceutical excipient that increases the aqueous solubility of Que, which is in good accordance with previously reported studies [105,106]. Results reported from in vitro analyses showed that P80-Que@Se had high water solubility compared to individual Que. In addition, P80-Que@Se NCs obtained low cytotoxicity in PC12 cell lines. Moreover, in vitro Cell Counting Kit (CCK) showed that P80-Que@Se could protect PC12 cells from H_2_O_2_-induced cell death. Furthermore, after the screening of the antioxidant effect with DPPH radical scavenging assay, it was reported that P80-Que@Se revealed high antioxidant activity. Based on the obtained results, it was concluded that this drug delivery method could serve as a useful outline for future studies on targeted drug brain delivery.

In order to improve the penetration of Que through the BBB, one other study was conducted where Que was loaded on solid lipid nano-particles (SLN) [107]. SLN of Que was formulated using a micro emulsification technique and after optimization of the formulations, based on physico-chemical properties (efficiency of entrapping Que inside the lipophilic core and ability of the SLN to release the entrapped Que, particle size, surface potential) SLN were selected for further in vivo studies in the Wistar rat model. After in vitro investigations, it was reported that the antioxidant activity of SLN showed an increase on the antioxidant activity leading to a conclusion that Que loaded SLN enhances brain delivery of Que. These findings indicate the vast potential SLN as a platform carrier to target various molecules to the brain, thus improving their efficacy in CNS diseases.

In the same line, Patil and his team in 2017 developed nano-structured lipid carriers of the Que for the nose-to-brain delivery, as a potential tool for targeted delivery. They have based their study on previously reported data, that led them in two directions: firstly, nose-to-brain delivery is the promising approach for the hydrophobic drugs since it has several advantages and increased stability [108] and secondly, colloidal carriers, due to their lipophilic properties, facilitate penetration via blood–brain and enter the brain by endocytosis [109]. Therefore, they prepared and characterized Que-loaded NLC. After the evaluation of physicochemical properties, penetration studies were performed ex vivo using nasal tissue from the sheep. In addition, nose-to-brain delivery studies were performed in vivo using Wistar rat models. Based on the reported results, Que NLC revealed sustained delivery of the drug and significant brain distribution was achieved in comparison to Que alone. Therefore, it was concluded that NLC might be a promising approach for the nose-to-brain delivery of Que.

Furthermore, aiming to develop a better platform for brain delivery of Que, Kumar et al., in 2016 focused their study on lipid nano-particles, particularly in Que loaded nano-lipidic carriers (NLC) and tested them on brain delivery. The resulting outcomes were compared with solid lipid nano-particles (SLN) [110]. The novelty of these formulations was involvement of lipid compositions which are a biocompatible and biodegradable composition of phospholipids, vitamin E acetate and glyceryl behenate used to enhance brain delivery. After optimization and characterization of Que encapsulation in nano-particles, followed by in vitro tests, it was reported that antioxidant properties were improved and cellular uptake increased in Caco-2 cell line, which confirms good intestinal permeability. In addition, the in vivo analysis was performed using rat animal model, for pharmacokinetic, biodistribution and brain delivery studies. Concerning pharmacokinetic studies, the differences in results were notable, where lipid nano-particles, particularly NLC significantly enhanced relative bioavailability (approximately six fold), biological residence (2.5 times) and substantially reduced drug clearance (approximately six fold). Consequently, both tested Que-nano-formulations enhance brain penetration leading to improvement in drug brain delivery. However, brain delivery for NLC was noticeably enhanced compared to SLN. Thus, based on the observations and reported results, the authors suggested SLN and NLC as a better vehicle for the brain delivery of Que [110].

In recent years, poly(n-butylcyanoacrylate) nano-particles (PBCA NP) have received considerable interest due to PBCA ability to overcome the limitation of other colloidal carriers [111]. PBCA has been widely investigated for targeted drug delivery due to its properties such as biodegradation, biocompatibility, bioadhesion and low toxicity [112,113]. The most remarkable findings of PBCA NP coated with polysorbate-80 (P-80) on their surfaces showed effectiveness in brain delivery since the particles entrap the compounds and prevent rapid elimination thus leading to enhancement through blood–brain penetration [114]. Therefore, Bagad et al. in 2015 based on these data, prepared Que—PBCA NP coated with P80 by the anionic emulsion polymerization method and investigated their physicochemical characteristics, in vitro release, stability, pharmacokinetics and biodistribution [115]. The observed results showed that Que was successfully entrapped using an anionic emulsion polymerization method (79.86% ± 0.45% and 74.58% ± 1.44% for QT-PBCA and QT-PBCA+P-80, respectively). Particles were spherical under TEM with an average size of 161.1 ± 0.44 nm (Que-PBCA NP) and 166.6 ± 0.33 nm (Que-PBCA NPs coated with P-80). In vitro release study of Que from Que-PBCA NP and Que-PBCA+P-80 NP showed a sustained release when compared with free Que. Besides, the relative bioavailability Que in Que-PBCA NPs and Que-PBCA+P-80 increased by 2.38- and 4.93-fold respectively, when compared to free Que. Furthermore, biodistribution tested in vivo in rat models, revealed that a higher concentration of Que was detected in the brain after the NP were coated with P-80. Based on the reported results, it can be indicated that PBCA NP coated with P-80 can be potential drug carriers for poorly water-soluble drugs such as Que. These NP improved the oral bioavailability of Que and enhanced its delivery to the brain.

In addition, aiming to develop an alternative carrier for Que that will result in bioavailability improvement, Wang and his team in 2016 used freeze-dried nano-micelles to load with Que [116]. Afterwards, the micelle characterization, release profile, cellular uptake, intracellular drug concentration, transport through the BBB, and antitumor efficiency were investigated. Cellular uptake and transport across the BBB were investigated in vivo, while the antitumor efficiency and distribution were evaluated in C6 glioma cell models. The observed results showed that Que-loaded freeze-dried nano-micelles had an efficient sustained release profile, increased intracellular uptake with low cytotoxicity, efficient penetration of BBB, and powerful cytotoxicity on C6 glioma cells. The authors suggested that freeze-drying micelles loaded with Que can be a promising delivery system for glioma treatment [116].

Moreover, many reports investigated liposomal formulations as a strategy for drug brain delivery. In general, studies have shown that the brain distribution of a drug that is loaded into a liposomal formulation depends on drug characteristics and liposomal formulation properties [117]. By previous findings, Priprem et al., in 2008 formulated Que liposomes and following the necessary physicochemical characterization they tested them on a rat animal model. They used two routes of administration (oral and intranasal) for further comparative studies. Based on the reported results, it was observed that intranasal Que liposomes showed a faster rate with a lower dose compared with oral administration of Que liposomes. Therefore, these results suggested that intranasal Que liposomes might be effective in the delivery of Que to the CNS [118].

A very interesting study that was recently reported was focused on developing nano-formulation that contains combination of two polyphenols: Que with curcumin [119]. Curcumin is another natural polyphenol that is derived from the spice—turmeric (Curcuma longa Linn, Zingiberaceae). Based on evidence reported from extensive research, it has been shown that curcumin exhibits potent antioxidant, anti-inflammatory and anticancer effects [120,121]. Therefore, based on previous data which support the hypothesis that the combination of Que and curcumin might have synergistic effects as anticancer agents, the current research develops nano-emulsion formulation with a combination of these two polyphenols for nose-to-brain delivery. Nano-emulsions with these two phytoconstituents (Curcumin-Que) were prepared with a 2:1 ratio. For preparation, the high-pressure homogenization technique was used. Further, nano-emulsions were characterized for drug content, globule size, zeta potential measurement, drug release and thermodynamic stability. The developed formulation was tested for cytotoxic activity on human glioblastoma U373-MG cells. In vitro examination for histological studies was performed on the isolated nasal mucosa of sheep, while in vivo studies were performed using an Allograft mouse tumor model. Based on the reported results, nano-emulsions with curcumin and Que (2:1) showed significant inhibition of the human glioblastoma U373MG cell growth. Therefore, from the reported results based on measurements from drug target efficiency and direct transport of drug from nose-to-brain, the effective CNS targeting via intranasal route has been noticed. Furthermore, based on an in vivo anticancer study in allograft mice models, the authors reported that the anticancer activity of the synergistic combination of these two polyphenols is higher as compared to doxorubicin. These interesting findings open a perspective in new directions for brain cancer treatment, initially showing that nano-emulsion increases nose-to-brain uptake, which is in good accordance with previously related reports [122], following with anticancer potential of the synergic combination of Que with curcumin. 

**Table 2 pharmaceutics-15-00963-t002:** Details of in vitro and animal studies of Novel formulations of Que and their effect on bioavailability and brain delivery.

Formulation	Cell Line/Animal Model	Main Findings on Que Bioavailability/Brain Targeting	References
Que-loaded with plasma exosomes	In vivo: AD Rats; I.V admin 12 mg/kg	(a) Improvement of pharmacokinetic parameters; (b) Que concentration increased by 2.5-fold in cerebrum and 1.5-fold in cerebellum (c) Biocompatible and safe	Qi et al., 2020 [123]
RVG29-Functionalized Lipid Nanoparticles for Que	In vitro: hCMEC/D3 cell line	(a) Increase 1.5-fold in the permeability across the in vitro model of the BBB; (b) With the highest concentration the cytotoxicity is lower than 15%	Pinheiro et al., in 2020 [89]
Transferrin- functionalized Lipid Nanoparticles for Que	In vitro: hCMEC/D3 cell line	(a) Increment in the permeability across the in vitro model of blood-brain-barrier; (b) No significant cytotoxicity in tested concentrations	Pinheiro et al., in 2020 [97]
Que-SPION	In vivo: healthy rat models	(a) Que concentration in plasma and brain are significantly higher (b) SPION does not cross BBB	Najafabadi et al., 2018 & Amanzadeh et al., 2019 [99,101]
Que SLN	In vivo: Wistar rat model	(a) Increased antioxidant activity (b) Enhanced brain delivery	Dhawan et al., 2011 [107]
Que NLC	Ex vivo: nasal tissue from the sheep; in vivo: Wistar rat models	Que NLC revealed sustained delivery of drug and significant brain distribution	Patil et al., 2017 [124]
(a) Que loaded nano lipidic carriers (NLC) composition of phospholipids, vitamin E acetate and glyceryl (b) Que loaded solid lipid nanoparticles (SLN)	In vitro: Caco-2 cell line—cellular uptake In vivo: Rat animal model—pharmacokinetic, biodistribution and brain delivery studies.	(a) Increased Que intestinal permeability; increased antioxidant activity (b) Enhanced bioavailability and brain penetration	Kumar et al., 2016 [110]
Que PBCA NP and Que PBCA NP coated with polysorbate-80	In vitro: (a) Drug stability (b) Release studies In vivo: Wistar rat model (a) pharmacokinetic studies (b) biodistribution	(a) Sustained release of Que (b) Relative bioavailability increased by 2.38- and 4.93-fold; (c) Increased concentration of Que in the brain with the Que PBCA NP coated with P-80	Bagad et al., 2015 [115]
SPION@APTES@FAPEG@ CQ; SPION functionalized with APTES ((3-Aminopropyl) triethoxysilane) and polyethylene glycol (PEG); conjugated with folic acid and loaded with Carboxylated Que	In vitro: Cytotoxicity- L929 cell line and U87 cells with MTT assay	SPION@APTES@FA- PEG did not show a cytotoxic effect; good ability for targeting cancer cells	Akal et al., 2016 [102]
Que-loaded freeze-dried nanomicelles	In vitro: C6 glioma cells and In vivo: BALB/c nude mice	Que-loaded freeze-dried nanomicelles had efficient sustained release profile, increased intracellular uptake with low cytotoxicity, efficient penetration of BBB, and powerful cytotoxicity on C6 glioma cells.	Wang et al., 2016 [116]
Que liposomes	In vitro: Rat animal model	Intranasal Que liposomes enhance brain delivery	Priprem et al., 2008 [118]
P80-Que@Se Que-loaded selenium nano-particles coated with acacia and P80;	In vitro: (a) cytotoxicity studies: PC12 cell line; (b) Cell Counting Kit (CCK) (c) DPPH radical scavenging assay –antioxidant activity	(a) Low cytotoxicity (b) Protection from H_2_O_2_-induced cell death (c) High antioxidant activity (d) Enhanced brain delivery	Yi et al., 2020 [103]
Nano-emulsion—nose-to-brain delivery Que and curcumin (2:1)	In vitro: (a) Cytotoxic effect—human glioblastoma U373-MG cell (b) Nose to Brain delivery -nasal mucosa of sheep In vivo: (c) Allograft mouse tumor model	(a) Significant inhibition of the human glioblastoma U373MG cell growth (b) Effective CNS targeting (c) Higher anticancer activity as compared to doxorubicin	Mahajan et al.,2021 [119]

RVG29, The peptide fragment of 29 amino acids from the rabies virus glycoprotein; hCMEC/D3, BBB model cell line; SPION, Supeparamagnetic iron oxide nano-particles; SLN, Solid Lipid Nano-particles, NLC, Nano-structured Lipid Carriers; P80, Polysorbate 80; DPPH, 2,2-diphenyl-1picryl-hydrazyl-hydrate.

## 5. Anticancer Activity of Que Nano-Delivery Systems: Evidence from In Vitro and In Vivo Studies

Considering the novel formulations of Que as a strategy to combat paediatric gliomas, numerous studies have been performed (Table 3) [7]. Since gliomas and glioblastoma have different metastatic potential and increased levels of drug resistance, their cell lines are the most frequently used in in vivo studies [125,126,127].

Ersoz et al. in their research synthesize and characterize nano-particles of Que loaded with poly (lactic-co-glycolic acid) with different sizes and encapsulation properties. Afterward, they evaluate their in vitro activity on C6 glioma cells and anticancer properties of different sizes of Que nano-particles on these cell lines. The method that was used for synthesis was single emulsion solvent evaporation. After the determination of properties such as particle size, zeta potential, polydispersity index and encapsulation efficiency, these formulations were tested on C6 glioma cells model. It was reported that all nano-particle formulations effectively inhibited cell proliferation. Additionally, it was shown that the formulations with the smallest size of nano-particles revealed better Que cellular uptake and antioxidant activity [128].

Similarly, the anticancer activity of Que in PEGylated drug delivery vehicle was investigated. Que nano-particles were coated with PEG2000-DPSE while tumor programmed cell death was studied on glioma C6 cells. Following the assay for cell survival, apoptosis, or necrosis, it was reported that Que coated with PEG2000-DPSE remarkably enhanced the anticancer effect of Que on C6 glioma cell lines [37].

Recently, in 2021, one of the most interesting studies regarding drug nano-delivery systems that target glioblastoma multiforme tumor, involves natural human platelets as a carrier for drug loading and drug delivery. The in vitro effect of Que and Que-platelet was evaluated on the U373-MG human astrocytoma glioblastoma cell line. Since the structure of platelets consists of an open canalicular system, it allows the uptake of Que molecules in platelet cytoplasm. The authors reported that in vitro at pH 5.5 that mimics the tumor microenvironment, the maximum encapsulation efficiency of Que-platelet was 93.96 ± 0.12% and that in 24 h the maximum drug release was 76.26 ± 0.13%. In addition, Que reached a threefold enhancement of solubility, followed by an increase of cytotoxic effect (after 48 h cell viability was 14.52%) [129].

Lou et al., in 2016, presented very promising results of Que nano-particles prepared with gold and loaded into PLGA in human neuroglioma cells U87. These cell lines were treated with Que nano-particles in different dosages, followed by subcutaneous injection in BALB/c-nude mice, aged 4–6 weeks. The tumor cell-inoculated mice were treated with Que nano-particles thorough intraperitoneal injection. Afterward, mice were sacrificed, and tumors were used for further analysis. Based on their results, authors concluded that Que nano-particles enhanced the inhibitory role in neuroglioma progression through multiple routes of action, mainly including autophagy and apoptosis. Autophagy was induced with increased conversion of microtubule-associated protein 1 light chain (LC3) from soluble form LC3-I to autophagosome-associated form LC3-II. Additionally, the accumulation of the p62 protein, which is considered an autophagy substrate protein, is decreased by inhibition of mTOR pathway which is regulated by suppression of PI3K/AKT signal. In a conclusion, Que loaded into PLGA nano-particles induced cell autophagy and apoptosis in human neuroglioma cells as well as suppressed tumor growth through activation LC3/ERK/Caspase 3 and suppression AKT/mTOR signaling [28].

Another study examined the cytotoxic effect of Que-loaded liposomes in C6 glioma cells [130]. The reported results initially indicate that Que-loaded liposomes increased Que solubility and improved bioactivity for inhibiting tumors. Furthermore, it was reported about the potential anticancer effects that might be the result of two routes of action. Primarily, the long exposure of C6 glioma cells with high Que concentration induced a reduction in glutathione levels and accumulation of highly reactive oxygen species level. Therefore, the pro-oxidant effect of Que could overcome the antioxidant effect which results in cell death. Moreover, the possible second anticancer effect mechanism could be through the effect of Que-loaded liposomes in increasing lactate dehydrogenase levels that is released during necrotic cell death (type III programmed cell deaths). In a conclusion, Que-loaded liposomes induce the enhancement of cytotoxic effects of Que in C6 glioma cells which appears as a result of type III (necrosis) programmed cell death [130].

Recently, Liu et al. in 2022 formulated Que-derived nano-particles that will target blood vessels of the brain tumor. They reported that after intravenous administration of previously synthetized Que-nanoparticles, Que binds to vessel in brain tumors, causing dual function—not only inhibition in the formation of new blood vessels but also the disruption of existing ones [131]. Besides Que, they formulated fifteen anti-angiogenic polyphenols into nanoparticles. These formulations were prepared using combinatory Fe-coordination and a polymer-stabilization approach. After the screening of anti-angiogenesis activity, they reported that nano-particles containing Que exhibit the greatest anti-angiogenic potential. Therefore, further investigation proceeded with Que nano-particles. Que nano-particles that were efficiently accumulated in GL261 glioma cells, intravenously were inoculated in mice models. From the presented results, the authors reported that Que nano-particles in mouse glioma models disrupted the existing tumor vasculature, leading to improved survival of tumor-bearing mice and enhanced drug delivery to a brain tumor. The anti-angiogenic mechanism that was proposed is through VEGFR2, which is overexpressed in tumor vessels. The authors demonstrated that the binding was mediated by Que, and the interaction of nano-particles with VEGFR2 leads to disruption of the existing tumor. In addition, previous data have shown that particular nano-particles can produce vessel disruption in tumors through intrusion on the homophilic interaction of vascular endothelial (VE)-cadherin, a phenomenon called “nanomaterials-induced endothelial leakiness” (NanoEL) [132,133]. In this study, it was reported that Que nano-particles induced NanoEL by selectively damaging the existing tumor vessels via VE-cadherin homophilic interaction disruption. These promising findings suggest that Que nano-particles as an anti-angiogenic agents have the potential to be used for brain cancer treatment, due to significant effect on reducing tumor development and enhancing drug delivery.

## 6. Synergism of Que with Other Therapeutic Agents

Vast amount of data showed that Que has potential in modulating cytotoxic cell responses to anti-cancer drugs by overcoming their therapeutic resistance. The efficacy of Que in brain tumors mainly in glioma and glioblastoma is supported with numerous reports of preclinical research. Several studies were focused on efficacy evaluation of Que combination with other therapeutic substances in order to verify synergistic potential of their particular combination. 

The synergistic effect of Que and temozolomide on human anaplastic astrocytoma cell line MOGGCCM was investigated by Jakubowicz-Gil et al. [33]. Their results indicated that Temozolomide induce autophagy, while Que promotes severe necrosis in the cell line, dependable from the applied concentrations of temozolomide. At a low concentration of temozolomide (5 ƞM), Que promotes the autophagic effect of drug, while at a higher temozolomide concentration (30 ƞM), autophagy switched to apoptosis. It was demonstrated that apoptosis was mediated by the mitochondrial pathway and the activation of caspase 3 and cytochrome C release, while no changes in caspase 8 expression were observed. Furthermore, it was reported that apoptosis was accompanied with inhibition of Hsp27 and Hsp72 expression and decreased mitochondrial membrane potential. Increased level of LC3II indicated the process of autophagy (Figure 5). It was also noticed that Temozolomide and Que changed the morphology of nucleus in MOGGCCM cells from a circular to an irregular shape. Based on the reported results, it was indicated that temozolomide administered with Que might be an effective and promising combination in glioma therapy.

Later on, the same team evaluated the apoptosis and autophagy effect of Que with temozolomide in human glioblastoma multiforme T98G cells [30]. Based on the observations, the authors reported that Que and temozolomide substantially stimulate apoptosis, while they have no effect on autophagy induction. Molecular mechanism is associated with activation of caspase-3 and 9 and with a mitochondrial membrane potential decrease. Moreover, both drugs are inhibitors of HSP27 and Hsp72 chaperones, that are involved in the prevention of cell death by a wide variety of agents that cause apoptosis (Figure 5). It was documented that the nuclear shape of the cells, was changed from circular form to “croissant like”. As a summary, these findings indicate that the apoptotic activity of temozolomide and Que in glioma multiforme T98G cells should be further elucidated on using this combination in clinical application.

Since it is well established that HSP27 chaperone is highly expressed in cancer cells by protecting malignant cells from undergoing apoptosis, it could be considered as a potential target for cancer treatment [134,135]. Sang and his team demonstrated in vitro the inhibitory effect of Que combined with temozolomide on HSP27 expression (Figure 5) [13]. They observed that temozolomide alone significantly increased Hsp27 phosphorylation in U251 and U87 cell lines, while in combination with Que, temozolomide induced Hsp27 phosphorylation and significantly inhibited Hsp27 expression. Moreover, it was reported that the synergy of Que with temozolomide showed a significant decrease in cell viability, when compared to temozolomide alone. In addition, it was observed that neither temozolomide nor Que affected caspase-3 activity and cell apoptosis, while their synergistic effect resulted in apoptosis and increased caspase-3 level. Based on the results, it was concluded that the Que sensitized glioblastoma cells to temozolomide by inhibiting HSP27 overexpression and increasing caspase-3 activity which leads to cell apoptosis and decreased survival of malignant cells.

Lately, one other study demonstrated that the co-administration of Que and antimalarial agent chloroquine in T98G, U251MG and U373MG glioma cells caused autophagy in these cell lines as a result of the excessive increase of autolysosomes and lysosomes, causing cell death. However, it is worth noting that no toxicity was observed in normal astrocytes (Figure 6) [14].

Preclinical research has shown that sodium butyrate, which is a histone deacetylase inhibitor, might have therapeutic potential in glioblastoma [136]. Taylor et al. demonstrated the effect of Que combined with sodium butyrate in rat C6 and human T98G glioblastoma cells (Figure 7). It was observed that this combination in treating rat C6 and human glioblastoma cells increased apoptosis by inhibition of protective autophagy [137].

Many well-documented reports showed that resveratrol and Que, which are structurally related and naturally occurring polyphenols, have anti-tumor effects [138,139,140,141]. Co-administration of these two polyphenols, in order to investigate their cytotoxic effect was tested on rat C6 glioma cells, human U87-MG and U138-MG glioma cells line (Figure 8) [142]. It was observed that the combination of resveratrol and Que triggered apoptosis, inhibited cell proliferation and the most significant activity was a strong synergism in inducing senescence-like growth arrest in gliomas cell lines. In this study, two types of effects were observed: high doses of resveratrol induced apoptosis, while low doses induced senescence and the co-treatment with low doses of resveratrol and Que induce both. Since these data suggest a synergistic effect of resveratrol and Que on senescence induction and long-term growth inhibition, it makes them suitable candidates for further preclinical studies on glioma tumors.

## 7. Que in Glioblastoma Multiforme—Case Study

Except the above-discussed studies conducted in vitro and in mice/rat models, regarding the effect of Que in brain cancer treatment, no other animal and human trials have been reported so far. However, one human case study was performed on an adult patient with advanced multifocal and rapidly progressing glioblastoma multiforme. Standard radiotherapy and temozolomide were combined with a single dose of 500 mg intravenously administered Que. In order to improve the BBB of Que, a leading-edge gamma knife was included in the treatment protocol. It was noticed that the patient had an improved ability to walk, balance and coordination and the requirements for steroid decreased. Consequently, patient life quality and response were improved compared to medical historical data. It has been proposed that these positive effects might come as a result of the inhibitory effect of the Que migration ability of glioblastoma cells [143].

## 8. Safety Aspects of Que Use

Based on an evaluation of scientific observations, Que is usually considered safe. According to existing reports, safety evaluation of Que usage was carried out in adult individuals. Children and adolescents as well as pregnant and breastfeeding women were not included in the target population. These specific vulnerable groups were excluded due to some safety concerns because of certain hormonal effects of Que that were observed in animal studies (variation of testosterone levels), which requires further elucidation [144]. In general, there is insufficient relevant safety data from human studies which could appeal to strong conclusions regarding safety concerns for this population group.

However, concerning the results that were obtained from animal studies but taking into consideration uncertainties due to limited human data and the challenges to extrapolating findings from animals to humans, particular potential risk groups have been identified. Patients with kidney malfunction might be a potential risk group for the long term Que usage at high doses based on the possible nephrotoxic effects of Que that were noticed on the pre-damaged kidney in rodents model [145,146,147]. In addition, in one animal study, it was observed that Que might have a potential tumor promotion activity in estrogen-dependent cancer [148]. However, these findings need further clarification.

Across several studies, adverse effects after Que intake have been rarely reported and such effects are usually mild. However, few reports indicated some potential risk groups. Therefore, the future investigation should be conducted in order to ensure the clinical safety of Que usage.

## 9. Conclusions and Remarks

Paediatric CNS tumors are the second most common childhood malignancy and the most common solid tumor in children. They are associated with significant morbidity and mortality. Over the last decade’s treatment protocols that include surgery, radiation and chemotherapy have improved outcomes in these patients. Current therapeutic strategies lead to high risk of side effects and therapeutic resistance. In this context, the use of flavonoids such as Que as discussed so far has surfaced as a promising alternative for brain tumors due to its antioxidant, anti-cancer and anti-inflammatory properties. Moreover, it is a natural compound with low toxicity and a suitable safety profile. Even though Que demonstrates significant antitumor activity, its application for further pharmacological benefits is limited due to its low solubility, bioavailability, and instability in different in vitro and in vivo experimental models.

Notably, the major breakthrough for Que is achieved via the encapsulation of Que into nano-formulations which improved its biological activity and therapeutic potential. However, so far there are no reported clinical trials employing nano-delivery systems for Que administration in CNS tumors. 

Several Que nano-formulations discussed in this review showed antitumor activity through different pathways. However, these experimental studies have been conducted on preclinical animal models and in vitro studies. Therefore, an important disadvantage is the insufficient knowledge of Que nano-formulation-associated risks in humans.

On the other hand, even though Que is considered generally safe, few studies indicated some potential risk groups after Que intake. Thus, additional research should be conducted in order to ensure safety aspects after Que administration.

As a conclusion, additional studies should be concentrated on the further development of Que nano-formulations as a targeted therapeutic option for brain cancers and those formulations should begin to be tested in humans following the investigation on animal models.

## 10. Future Perspectives

The global initiative for research and development of nano-medicine has been growing firmly, mainly focused on anticancer agents [149]. In this context, nano-medicine appears is a promising tool for optimized paediatric therapy [150]. So far, several research institutions started with pioneering projects on research and development of nano-paediatrics [151,152]. Some of the government-promoted initiatives in the US and Europe, i.e., the Paediatric Investigation Plan (PIP), might improve advances in paediatric nano-medicine research [153,154,155]. In addition, European Medicines Agency (EMA) created the Paediatric Committee Formulation Working Group, that can be considered as an open path to concretely implement nano-technology, in order to promote the development of innovative paediatric nano-formulations [156,157].

Apart from these and other regulatory initiatives, the outcome for paediatric nano-medicines is strongly dependable on the progresses made in the development and implementation of these formulations in adults. In addition, a persistent challenge remains the absence of a regulatory framework for nano-medicines resulting in prolongation of implementation of nano-technology in the paediatric population even after long and extensive research [28].

Meanwhile, paediatric brain tumors remain one of the leading causes of childhood morbidity and mortality. An improved understanding of the drug resistance will certainly open new insights for the development of targeted therapies, boosting the cure rates further.

Alternative therapies based on natural components which usually derive from plant matrixes, might be suggested as potential directions for future research in this field. Flavonoid Que, due to its relevance in preclinical anti-cancer activity shown in adult research models, which was additionally enhanced using nano-technology, was discussed in detail in this review. 

Taken altogether, as Que nano-formulations are being regarded to have a considerable anti-cancer potential, further studies may lead to new paths for paediatric brain cancer treatments.

## Figures and Tables

**Figure 1 pharmaceutics-15-00963-f001:**
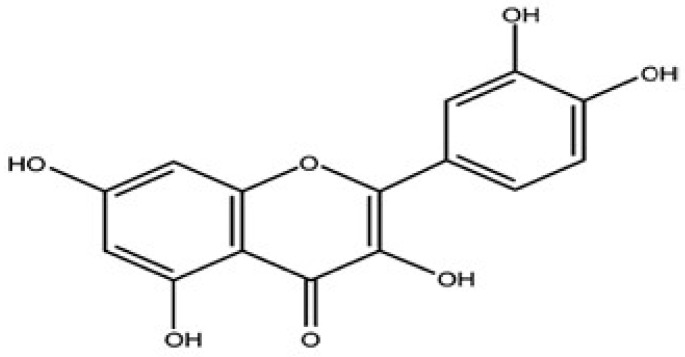
Structure of Que.

**Figure 2 pharmaceutics-15-00963-f002:**
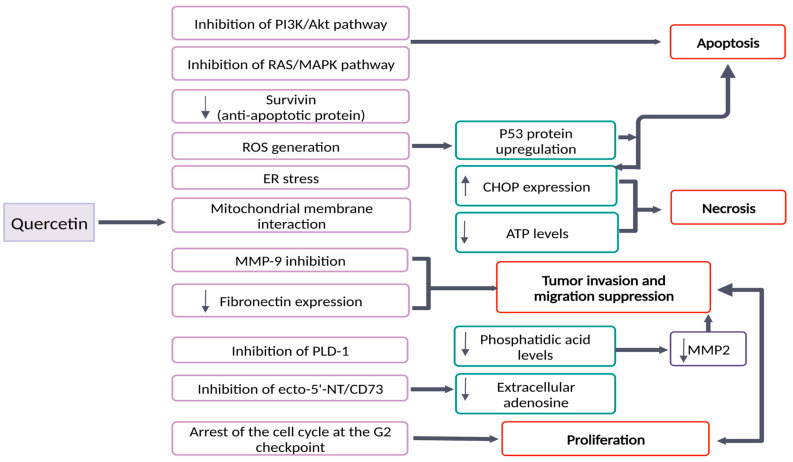
Some of the possible targets that are related to quercetin anticancer activity; (created with www.biorender.com; 2 March 2023); ↑ Increase; ↓ Decrease.

**Figure 3 pharmaceutics-15-00963-f003:**
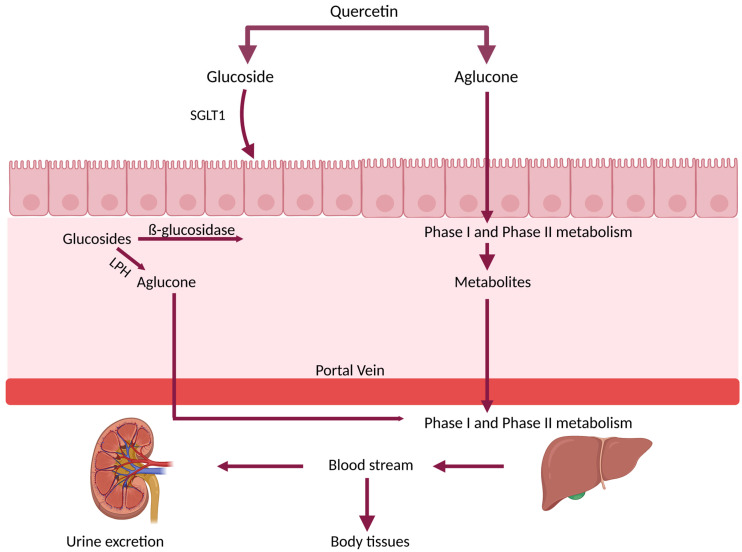
General overview of Que and Que glucoside bioavailability. Abbreviations: SGLT 1, sodium-dependent glucose transporter; LPH, lactase-phlorizin hydrolase enzyme (created with www.biorender.com; 25 February 2023).

**Figure 4 pharmaceutics-15-00963-f004:**
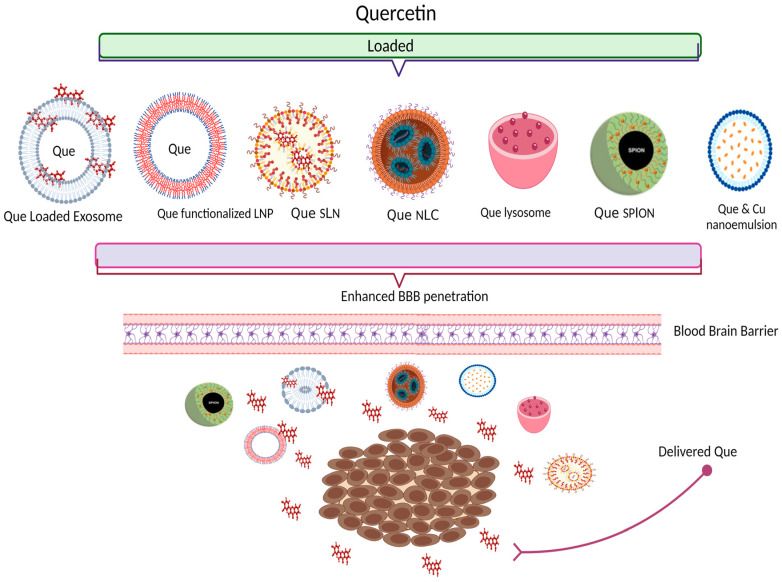
Schematic presentation of Que different nano-delivery systems that facilitate BBB penetration and target-specific action in glioma cells. Abbreviations: LNP, Lipid nano-particles; SLN, solid lipid nano-particles; NLC, Nano-structured lipid carriers; SPION, Superparamagnetic iron oxide nano-particles; Cu, Curcumin; (created with www.biorender.com; 25 February 2023).

**Figure 5 pharmaceutics-15-00963-f005:**
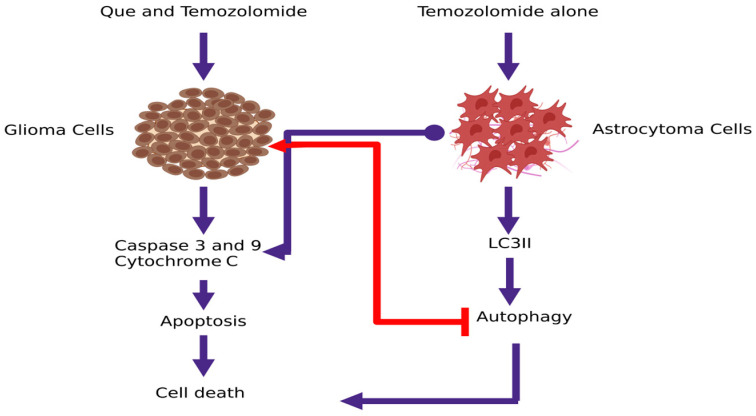
Synergistic cytotoxic effects of Que and temozolomide tested on human glioblastoma multiforme T98G cells and human anaplastic astrocytoma cell line MOGGCCM. Que synergistically enhances temozolomide-mediated apoptosis by cytochrome C release, activation of caspase 3 and 9 and inhibition of Hsp27 and Hsp72 expression that are involved in the prevention of cell death [30]. At a low concentration of temozolomide, Que promotes autophagic effect that indicate an increased level of LC3II, while at a higher temozolomide concentration, autophagy switched to apoptosis [33]; (created with www.biorender.com; 25 February 2023).

**Figure 6 pharmaceutics-15-00963-f006:**
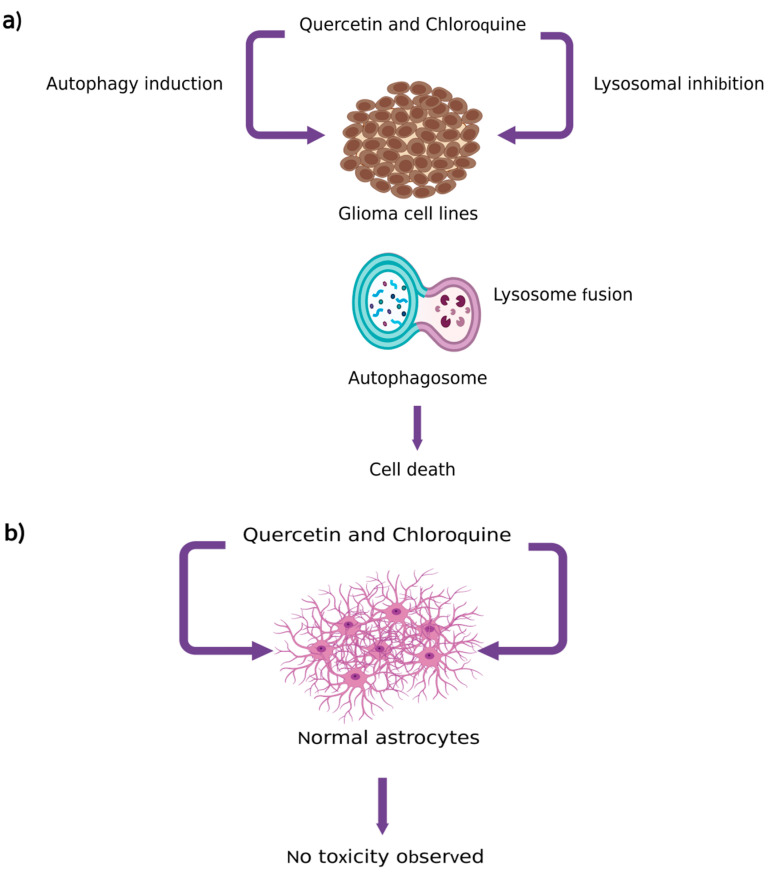
Co-administration of Que with chloroquine in (**a**) glioma cell lines and (**b**) normal astrocytes cells; (**a**) Que synergistically enhanced chloroquine-mediated lysosomal inhibition and triggered autophagy leading to excessive increase of autolysosomes and lysosomes causing cell death; (**b**) no toxicity effect resulted after co-administration of Que with chloroquine in normal astrocyte [100] (created with biorender.com; 25 February 2023).

**Figure 7 pharmaceutics-15-00963-f007:**
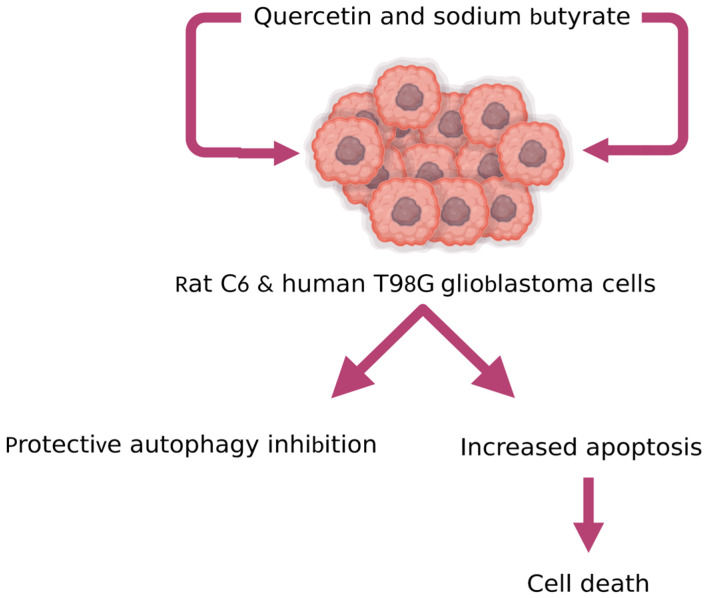
The combination of Que with sodium butyrate in rat C6 and human glioblastoma cells increased apoptosis by inhibition of protective autophagy [102] (created with www.biorender.com; 26 February 2023).

**Figure 8 pharmaceutics-15-00963-f008:**
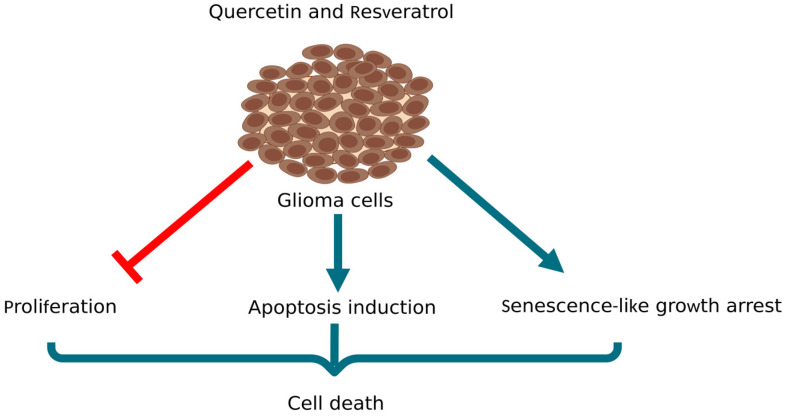
Co-administration of Que with resveratrol in rat C6 glioma cells, human U87-MG, and U138-MG glioma cells line inhibited cell proliferation, triggered apoptosis and induced senescence-like growth arrest [107] (created with www.biorender.com; 26 February 2023).

**Table 3 pharmaceutics-15-00963-t003:** Novel formulations of Que and their effect on glioma and glioblastoma cell lines.

Modified Que Formulations/Que with Other Therapeutic Molecules	Cell Line/Animal Model	Main Findings	References
Que-loaded nano-particles with poly (lactic-co-glycolic) acid (PLGA)	C6 glioma cells	Que-loaded nano-particles with poly (lactic-co-glycolic) acid (PLGA) effectively inhibited the cell proliferation. The smallest size of nano-particles revealed the better Que cellular uptake and antioxidant activity.	Ersoz et al., 2020 [128]
Que coated with PEG2000-DPSE	C6 glioma cells	Que coated with PEG2000-DPSE enhanced anticancer effect on C6 glioma cell lines	Wang et al., 2013 [37]
Que-platelet (Natural human platelets)	U373-MG cell line	Que obtained threefold enhancement of solubility, and increasement of cytotoxic effect.	Bhandarkar et al., 2021 [129]
Que nano-particles prepared with gold and loaded into PLGA	Human neuroglioma cells U8 and BALB/c-nude mice	Induction of cell autophagy and apoptosis in human neuroglioma cells as well as suppressed tumor growth through activation LC3/ERK/Caspase 3 and suppression AKT/mTOR signaling	Lou et al., 2016 [28]
Que-nano-liposomes	C6 glioma cells	Que-loaded liposomes has enhanced cytotoxic effects inducing type III programmed cell death (necrosis) in C6 glioma cells.	Wang et al., 2012 [130]
Que-nano-particles	Mouse model	Que nano-particles via anti-angiogenic have significant effect on reducing tumor development and enhancing drug delivery.	Liu et al., 2022 [131]

U373 MG, human glioblastoma cell lines; PLGA, Poly (lactic-co-glycolic acid).

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
