# Peer review of "Quercetin and Its Nano-Formulations for Brain Tumor Therapy—Current Developments and Future Perspectives for Paediatric Studies"

_pharmaceutics, 2023, doi:10.3390/pharmaceutics15030963_

Round 1

Reviewer 1 Report

The review written by Shala et. al. “Quercetin and its modified formulations on pediatric brain tumors – current developments and future perspective” is comprehensive and mostly well written. However, the figures are quite distorted and disorganized. Hence, to enhance its quality and presentation, the manuscript should be thoroughly revised based on the following suggestions and submitted back for reevaluation and consideration for publication.

1.      Please add some more references along with reference 1 for the first paragraph of the introduction.

2.      Also, please add references to the sentence having line no. 37-39 and 82-83.

3.      Research all round the world indicate environmental risk factors as one of the key players in various types of malignancy. Cite a few references related to their role in brain tumor formation.

4.      In line 98 please use the full form of “Que” for the first time before introducing the abbreviated form.

5.      The authors describe the molecular mechanisms of action of quercetin in gliomas and glioblastoma ccell lines, however the studies they highlighted didn’t mention the cell lines used in respective studie to be of pediatric origin.  Therefore, the sentence “The most common types of tumors in children, gliomas and glioblastomas, were employed in all of the studies described above” having line numbers 203-204 seems to be confusing and needs to be rewritten so that it’s not misinterpreted.

6.      Also, in context with the previous comment, line 125-153 has detailed description of the mechanism of apoptosis but is under the subheading “Quercetin’s effects in brain tumors: Molecular mechanisms and signaling pathways” which doesn’t seem very relevant . Primarily, I don’t feel the relevance of such an elaborate discussion on apoptosis, however if the authors think it’s important, they can move this part somewhere more contextual and make it a separate paragraph with different heading.

7.      In line 231 “However, studies examining the relative bioavailability of flavonoids in general are more common” the authors mention “studies” but cites only one reference. Please correct and cite more articles.

8.      In Figure3 it seems like the pictures of liver, blood cells and kidney are downloaded from the internet and then modified a bit. It is very surprising that they didn’t acknowledge the source thereafter. This is not allowed in scientific writings at all. The authors should redraw all the figures/ pictures of organs by themselves and reconstruct the figure accordingly. Also, they should choose subtle colors as background when writing above it, not like the dark red used for the boxes blood stream, metabolites and portal vein.

9.      For Figure4, some of the nanodelivery system pictures seems distorted especially when they are depicted to target the glioma cells. Its also very difficult to understand the labeling inside the Que plasma exosome. Please revise the figures accordingly.

10.   The glioma cell line picture shown in figure 6A and figure7 seems to be inaccurate flattened version of the glioma drawn in Figure5. Also, in figure 6A, the autophagosome and lysosome fusion seems to be hazy, and internet downloaded picture. Please make sure that all the figures or pictures used in them are clear, self-made, free from apparent distortions. This applies for figure 7 and 8 also, having similar issues.

11.   In Figure 6B, what is the significance of using 2 color to depict astrocytes, explain?

12.   A summarized figure containing the advances in pediatric nanomedicine so far and the future perspectives of this article can enhance the rigor of the review article.

Author Response

Please, consider the attached file.

Reviewer 2 Report

In the present review, the authors try to discuss the antitoxicity effects of quercetin and its formulations for treating pediatric brain tumors. Since investigations about quercetin and pediatric brain tumors are not enough but quite needed, it’s nice to summarize them in a review for researchers to read. However, the text is not well arranged and the logic of some paragraphs is not clear.

 For example, the authors discuss the role of bioavailability and nanotechnology which is not required to be extensively discussed. I think, the review is not insightful and does not relate to just pediatric tumors, this is the reason the authors keep switching to the usage of CNS cancers now and then through the manuscript. In that case, the authors should have done a fair job in putting proper specific claims forward in their manuscript. They failed to conceptualize on the fact that quercetin is a polyphenol and could have used it as a basis to start their manuscript and include it in the introduction. It would have been better to build the article around quercetin than pediatric tumors because then, quercetin has better and wider literature.

 Further, there also concerns such as in table 1 there is no cross referencing as compared to other tables. The authors have not italicized “In vitro” or “in vivo”. I would also suggest to add a row below each table highlighting the abbreviations used in the table or at the end in a subsection or before the usage of the word followed. Please also use the abbreviated form “Que” throughout the manuscript. In some places the word quercetin has been written throughout. I believe the authors have also not used numbering for their subsection. This showcases the poor plight in editing and submitting to this journal.

Overall, I think the manuscript needs a major overhaul and restructuring to give critical/insightful/intuitive comments on the title of the manuscript.  Besides, there are already manuscripts available on the similar lines as the one presented here. I wonder, what is the novelty then?

Author Response

Please, consider the attached file.

Reviewer 3 Report

This is a work that reviews both past and recent efforts that employed quercetin for brain tumor treatment. However, this reviewer fails to identify the pediatric aspect in all the presented studies. This should probably be given as a perspective in this article, rather as the main concept. Some comments:

-The title must be improved. I do not agree with the term "modified formulations", though I understand it refers to modified release.
-English is mostly ok, but requires thorough checking and improvement throughout the manuscript.
-Intro is confusing to me, going back and forth on tumor issues in children and not addressing the review topic at all. Maybe elements from the next parts should be incorporated into the intro somehow.
-Part 2 is also confusing and should be revised.
-Table 1
--Author/Year and Study Title columns should be replaced by a Reference column at the right end of the table.
--The meaning of the arrows should be explained below the table and their position next to the results should be fixed properly.
--The first column could be the utilized formulation of quercetin, whether a classic pharmaceutical form or a nanosystem.
-Figure 4 the included abbreviations should be explained.
-Table 3 (should be named Table 2)
--Same with Table 1
-Part on QUE nanoparticles should be better organized. For example, SPIONS are mentioned at the beginning and then later they are explained again.
-Part on novel DDSs does not explain the distinctive characteristic of these systems compared with nanoparticles presented before
-Table 2 (should be named Table 3)
--Same with Table 1
-Statement of figure adoption should be mentioned whenever applicable.
-Image quality is generally not satisfactory.

Author Response

Please, consider the attached file.

Round 2

Reviewer 1 Report

Authors have addressed the comments well and revised the manuscript accordingly. It looks good to be accepted for publication in its present revised format.

Author Response

Dear Editor,

We are writing in response to the feedback provided on our manuscript. Initially, we would like to express our gratitude for the time and effort you have put into evaluating our work and providing valuable feedback to improve the quality of the manuscript.

In this revised version, we have considered all the comments and suggestions provided by the reviewers and have made the necessary changes to the manuscript. We believe that our revised manuscript now better addresses the concerns raised by the reviewers while maintaining the integrity and clarity of the work.

Please find below the detailed response to the comments, highlighting the changes made to the manuscript and addressing any concerns or queries that were raised. We hope that the revised manuscript and our response adequately address all of the reviewers' concerns and comments.

Thank you once again for your invaluable feedback, and we look forward to hearing back from you soon.

Reviewer 1

  1. Authors have addressed the comments well and revised the manuscript accordingly. It looks good to be accepted for publication in its present revised format.
  • We want to express our deepest gratitude for the reviewer invaluable contribution in the revision process of our manuscript. His/her thorough feedback and constructive criticism had significant role in helping us improve the quality of the manuscript.

Reviewer 2 Report

In the heading of Table 2, the authors have not italicized "in vitro"

Author Response

Dear Editor,

We are writing in response to the feedback provided on our manuscript. Initially, we would like to express our gratitude for the time and effort you have put into evaluating our work and providing valuable feedback to improve the quality of the manuscript.

In this revised version, we have considered all the comments and suggestions provided by the reviewers and have made the necessary changes to the manuscript. We believe that our revised manuscript now better addresses the concerns raised by the reviewers while maintaining the integrity and clarity of the work.

Please find below the detailed response to the comments, highlighting the changes made to the manuscript and addressing any concerns or queries that were raised. We hope that the revised manuscript and our response adequately address all of the reviewers' concerns and comments.

Thank you once again for your invaluable feedback, and we look forward to hearing back from you soon.

Reviewer 2

  1. In the heading of Table 2, the authors have not italicized "in vitro"

Initially, we want to thank the reviewer for time and effort in reviewing our manuscript and for providing valuable feedback. The detailed feedback and helpful criticism provided by him/her played a significant role in enhancing the manuscript's overall quality.

  • Accordingly with reviewer comment, in vitro is italicized in the heading of Table 2 (Page 25; line nr 696 is corrected)

Reviewer 3 Report

This reviewer still believes that the matter of pediatric tumors has not been successfully correlated with quercetin and that a perspective discussing the possible use of the molecule as such does not justify the title.

It is not customary and practical to include study titles inside a table. I suggest removing them from all tables and in the case of Table 1, replacing it with something else.

This reviewer meant that there is no explanation on how systems from chapter 5 differ from those from chapter 4. Please provide an explanation.

Author Response

Dear Editor,

We are writing in response to the feedback provided on our manuscript. Initially, we would like to express our gratitude for the time and effort you have put into evaluating our work and providing valuable feedback to improve the quality of the manuscript.

In this revised version, we have considered all the comments and suggestions provided by the reviewers and have made the necessary changes to the manuscript. We believe that our revised manuscript now better addresses the concerns raised by the reviewers while maintaining the integrity and clarity of the work.

Please find below the detailed response to the comments, highlighting the changes made to the manuscript and addressing any concerns or queries that were raised. We hope that the revised manuscript and our response adequately address all of the reviewers' concerns and comments.

Thank you once again for your invaluable feedback, and we look forward to hearing back from you soon.

Reviewer 3

We would like to express our gratitude to the reviewer for dedicating their time and effort to review our manuscript and for sharing valuable feedback. Their detailed feedback and constructive criticism greatly contributed to improving the overall quality of the manuscript.

1.This reviewer still believes that the matter of pediatric tumors has not been successfully correlated with quercetin and that a perspective discussing the possible use of the molecule as such does not justify the title.

  • We understand the reviewer’s point of view. Compared to the first version of the manuscript, the review was reorganized by focusing on quercetin and its use in the treatment of brain cancers. The aim of this review is to describe recent research and discoveries regarding the anticancer potential of quercetin in brain tumors. In particular, our review reports the potential role of quercetin as anti-cancer drug, based on rigorous pre-clinical research studies. Although there is only one case report on a human subject, the lack of any conducted clinical studies to date does not diminish the importance of the reported data. Therefore, we strongly recommend further clinical research studies to investigate the anti-cancer activity of quercetin, given the compelling evidence presented in this review. Further studies in the paediatric setting are also recommended, as all published studies indicate the antitumor potential of quercetin used in adult models. For this reason, we modified part of the title from "future perspectives in paediatric treatments" to "future perspectives in paediatric studies".
  1. It is not customary and practical to include study titles inside a table. I suggest removing them from all tables and in the case of Table 1, replacing it with something else.
  • Accordingly with reviewer suggestion titles from the tables are removed, while column at table 1 was replaced as well.

  1. This reviewer meant that there is no explanation on how systems from chapter 5 differ from those from chapter 4. Please provide an explanation.
  • Section 4 presents the findings on the utilization of Que nanodelivery systems to overcome its bioavailability and brain delivery limitations, thereby enhancing its effectiveness in targeting specific tumor cells. This section also discusses the role of Que nanodelivery in circumventing the obstacles posed by the BBB, which restricts the delivery of Que to the brain.
  • Section 5 is focused on reported data that indicates the anticancer activity of Que nanoparticles. In addition, the study that is focused more on bioavailability advantages of Que nanoparticles is removed from section 5 and added on Section 4 (Page 35; Lines 763-773 are removed and added to page 24; lines 628-638). Accordingly, the same reference is removed from Table 3 and added on Table 2 (Wang et al., [85]). Moreover, in order to avoid confusion, the title of Subsection 5 has been changed from: “Novel drug delivery systems of Que – in vitro and in vivo studies” to “Anticancer activity of Que nanodelivery systems: evidence from in Vitro and in Vivo studies” (Page 34; line 703-704).